# Integrated Management of Constipation in Hypothyroidism: Evaluating Pharmacological and Non-Pharmacological Interventions

**DOI:** 10.3390/nursrep15100354

**Published:** 2025-09-29

**Authors:** Eman M. Gaber Hassan, Sharell Lewis, Sajedah Fawzi Alsadiq, Salha Ali Almarhoon, Hanan Mufareh Alsubeh, Sana Mohammad Alboori, Khulood Abdulghafour Al Marzooq, Fatimah Saleh Al Awami, Mohammad Daud Ali

**Affiliations:** 1Department of Nursing, Mohammed Al-Mana College for Medical Sciences, Dammam 34222, Saudi Arabia; emanm@machs.edu.sa (E.M.G.H.); sharelll@machs.edu.sa (S.L.); 2Medical-Surgical Nursing Department, Faculty of Nursing, Cairo University, Cairo 11562, Egypt; 3Intensive Care Unit (ICU), Qatif Central Hospital, Eastern Health Cluster, Al Qatif 32654, Saudi Arabia; sfalsadiq@moh.gov.sa; 4Home Health Care, Eastern Health Cluster, Zone D, 4696 King Abdullah, Jubail City Center, Al Jubail 32253, Saudi Arabia; saalalmarhoon@moh.gov.sa; 5Urgent Care Clinic (UCC), Bader 91 PHC, Eastern Health Cluster, Dammam 32265, Saudi Arabia; halsubeh@moh.gov.sa; 6Pediatric Ward, Maternity and Children Hospital, Eastern Health Cluster, Dammam 32253, Saudi Arabia; smalboori@moh.gov.sa; 7Nursing Office, Dammam Medical Complex, Eastern Health Cluster, Dammam 32253, Saudi Arabia; kaalmrzooq@moh.gov.sa; 8Clinical Nurse Educator, Nursing Department, Safwa General Hospital, Safwa 32827, Saudi Arabia; fasalawami@moh.gov.sa; 9Department of Pharmacy, Mohammed Al-Mana College for Medical Sciences, Dammam 34222, Saudi Arabia

**Keywords:** chronic constipation, hypothyroidism, pharmacological approaches, non-pharmacological approaches, patient-centered care, chronic care

## Abstract

**Background/Objective:** Chronic constipation is a common gastrointestinal disorder that can be caused by a variety of factors, such as demographic, lifestyle, and medical disorders like hypothyroidism. Its prevalence varies worldwide, affecting quality of life and leading to specialized management strategies. To explore hypothyroidism patients’ knowledge and practice regarding constipation and evaluate the perceived effectiveness of pharmacological and non-pharmacological management approaches. **Methods:** A descriptive, cross-sectional design was used to collect the data from a private hospital in the eastern region of Saudi Arabia from January to May 2025. A convenient sample of 300 individuals with hypothyroidism completed the Bowel Habits Questionnaire. **Results:** Most participants knew that hypothyroidism could cause constipation, but they reported that they did not have more knowledge about it. Both pharmacological and non-pharmacological interventions, especially increase water intake, fiber intake, and exercise, were commonly used by the participants, and they perceived these approaches to be effective. There were strong correlations between constipation frequency and age, disease duration, and the use of constipation management methods. A strong association was found between constipation management strategies and treatment effectiveness. **Conclusion:** Age, disease duration, and constipation management strategies significantly affect constipation in hypothyroidism patients. Drinking plenty of water and eating more fiber are two very effective non-pharmacological strategies. It is recommended that nurses who integrate routine bowel health education and lifestyle guidance into care plans consider the gap in patient knowledge regarding the relationship between hypothyroidism and constipation, to enhance patients’ self-management and contribute to better health outcomes.

## 1. Introduction

Constipation is a common gastrointestinal disorder characterized by infrequent bowel activity, difficulty passing stools, hard stool consistency, bloating, or a persistent sense of incomplete evacuation, and in some cases, individuals may require manual assistance to defecate [1,2]. It has been defined clinically as having less than three bowel movements per week [3]. According to the American College of Gastroenterology, chronic constipation is diagnosed when these symptoms persist for at least three months and include prolonged straining or delayed passage of stool [4,5]. This difficulty may involve excessive straining or prolonged defecation time [6]. However, definitions can differ depending on patient experiences, physician assessments, cultural influences, and changing clinical criteria [4].

The prevalence of chronic constipation globally shows considerable variation. In Western populations, Canada, reported rates ranges from 2% to 27%, while in the United States, this condition affects about 16% of individuals [7,8]. Across Asia, the prevalence rate varies—about 8% in China, 17% in India, and up to 28% in Japan [9,10,11]. A comprehensive global review analyzing data from 45 studies and over 275,000 participants reported functional constipation rates at 15.3% using Rome I criteria and 10.1% using Rome IV [12]. Within Saudi Arabia, the prevalence is remarkably diverse. For instance, Alhusainy et al. (2018) documented rates ranging from 25% to 60% in Riyadh depending on the diagnostic method used, while Alhassan et al. (2019) found significant disparities in constipation prevalence among Saudi cities, with 83.3% in Riyadh compared to 16.7% in Qassim [13,14].

Chronic constipation is a complex condition influenced by various demographic, socioeconomic, lifestyle, and health-related factors. Research has shown correlations with age, gender (especially among females), educational attainment, income level, marital status, and place of residence, such as living in urban or rural areas [15,16]. A study found that certain lifestyle habits, such as a lack of adequate physical activity, not drinking enough fluids and dietary fiber, smoking, drinking alcohol, and taking medications, can also contribute to the condition [17]. Additionally, a variety of medical conditions affecting systems such as the gastrointestinal, neurological, and endocrine play a significant role [15,16]. Hypothyroidism is especially associated with constipation due to it slowing down gastrointestinal movement. This may be a result of neuromuscular dysfunction, hormonal resistance, and changes in the structure of the intestinal wall, such as thickening of the muscular layer and shortening of intestinal villi [18,19,20,21,22]. Physiological studies have reported that thyroid disorders can affect anorectal function and intestinal transit, but the symptoms can vary between individuals with normal, low, or high thyroid function [23,24,25,26]. Hypothyroidism affects about 13.9% of the people around the world, with higher rates reported among older adults and women, especially in Eastern and Southern Europe [27,28,29]. In Saudi Arabia, the prevalence rate is between 18.7% and 29.1%, and women were diagnosed at a much higher rate [30,31].

Non-pharmacological interventions for managing chronic constipation are common, especially among patients with mild symptoms. The management may include recommendations such as lifestyle modifications that focus on increasing physical activity, increasing fluid intake, consuming a high-fiber diet, and establishing regular bowel habits [32,33]. In order to optimize the bowel function, patients are generally advised to drink 1.5 to 2 L of water on a daily basis, as well as consume 20–35 g of fiber [34]. To meet these fiber requirements, the patients can source them naturally from the diet or through supplements such as psyllium, which not only improves bowel regularity but also supports glucose control and gut microbiota balance [33]. In cases wherein pelvic floor dysfunction is involved, approaches such as bowel training and biofeedback have shown significant benefits [34]. The most essential components of long-term management include educating patients on the importance of sustainable healthy habits and promoting physical activity [35,36]. Traditional Chinese Medicine and acupuncture, which are complementary therapies, have shown promise in regulating gut flora; however, additional evidence is needed to confirm their effectiveness [37]. The replacement of thyroid hormone remains a key strategy to improve constipation by normalizing gastrointestinal motility among hypothyroidism patients [38]. The above-mentioned approaches may serve as adjuncts to conventional management. A stepwise approach to pharmacological options becomes necessary when lifestyle changes alone are insufficient for managing constipation.

Bulk-forming agents, such as psyllium, polycarbophil, and methylcellulose, are typically used as a first-line treatment. If needed, osmotic laxatives such as polyethene glycol (PEG), lactulose, or sorbitol are next used [32,34,39]. Since stimulant laxatives like senna, bisacodyl, or sodium picosulfate are considered effective for many patients, they are introduced when treatments appear not to be working [40,41,42,43]. For patients with refractory symptoms, advanced therapies, such as secretagogues (linaclotide, lubiprostone), 5-HT4 agonists like prucalopride, and magnesium-containing compounds, are recommended [5,35]. In some cases, combining different agents such as PEG with bisacodyl or magnesium oxide with sodium picosulfate and citric acid can produce faster and more effective results [4]. Surgical options are considered only for patients who do not respond to conservative and medical interventions [36,40]. Ultimately, effective management must be tailored to the patient’s specific condition, underlying causes, and overall health profile to achieve the best outcomes.

Despite the well-acknowledged link between hypothyroidism and constipation, little research has been done in Saudi Arabia that focuses on the treatment of constipation in hypothyroidism patients. Most available Saudi studies focused on the prevalence of hypothyroid symptoms or general constipation, evaluating the effectiveness of the constipation treatment strategies.

This study addresses that gap by examining what hypothyroid patients know about constipation and how they manage it, whether through pharmacological and non-pharmacological interventions or both. The goal is to gain a deeper understanding of current practices and improve patient care. The aim of the study is to explore hypothyroidism patients’ knowledge and practice regarding constipation and evaluate the perceived effectiveness of pharmacological and non-pharmacological management approaches.

## 2. Methods

### 2.1. Design and Setting

A descriptive, cross-sectional research design was used to evaluate the management of constipation among adults diagnosed with hypothyroidism and being treated in private hospitals in the eastern region of Saudi Arabia from January to May 2025.

### 2.2. Sample

A non-probability convenience sample was used to collect information from patients with hypothyroidism from outpatient clinics. The recruitment is specifically conducted during fac-to-face encounters in outpatient endocrinology and internal medicine clinics within these hospitals. According to national estimates, the prevalence of hypothyroidism in Saudi Arabia is approximately 49.76% [44]. The Saudi eastern province which has a total population of approximately 4,900,325, include an estimated 5.3% (245,017 individuals) diagnosed with hypothyroidism. The Raosoft sample size calculator (http://www.raosoft.com/samplesize.html) (accessed on 1 January 2025) was used to calculate the sample size. The estimated number of participants needed is 380, with a 5% margin of error and a 95% confidence interval [45]. The inclusion criteria for this study targeted adults 18 years and older with a confirmed medical diagnosis of hypothyroidism for at least 6 months to reduce variability in symptom fluctuation due to recent treatment changes. The participants should be stable on thyroid medication and were receiving care in outpatient hospital clinics in the Eastern Province of Saudi Arabia. Eligible participants were required to have experienced symptoms of constipation for at least three consecutive months, consistent with the clinical definition of chronic constipation. Both Saudi citizens and non-Saudi individuals were included to ensure diversity in the sample. Additional criteria included the ability to provide informed consent, adequate cognitive capacity to complete the questionnaire, and welling to participate voluntarily.

In contrast, individuals were excluded from the study if they were younger than 18 years, patient with no prior history of constipation, and those whose constipation was attributable to non-thyroidal causes, such as irritable bowel syndrome, inflammatory bowel disorders, gastrointestinal disorders, colon cancer, etc. Exclusion criteria, also applied to those who had recently undergone abdominal or pelvic surgery and were taking medications that markedly influence bowel motility (including opioids, anticholinergic, iron supplements, and certain antidepressants). Finally, individuals with severe psychiatric illness or significant cognitive impairments that could interfere with reliable data collection were not eligible to participate.

### 2.3. Study Tools

The research utilized a structured questionnaire, The Bowel Habits Questionnaire, to evaluate hypothyroid patients’ knowledge, attitude, and practice regarding bowel complications associated with their condition, with a focus on constipation. The tool also evaluates the pharmacological and non-pharmacological interventions patients use to manage constipation. The questionnaire was developed by the researcher following an extensive review of the relevant literature to ensure alignment with the study objectives and existing knowledge on hypothyroidism and chronic constipation.

The questionnaire consists of four parts, each part in the questionnaire was structured with either closed-ended or multiple-choice questions or a Likert scale to gather comprehensive insights:Part 1: Demographic Information, which assesses basic participants’ information (such as age, gender, duration of hypothyroidism diagnosis, and chronic disease). These variables were essential to analyze association between patient profiles and constipation management.Part 2: Participants’ Awareness of Bowel Complications Associated with Hypothyroidism: this section assessed patients’ awareness about the impact of hypothyroidism on bowel function. It explored whether participants recognized constipation as a complication of hypothyroidism, whether they had discussed bowel habits with their physician, and their satisfaction with current management strategies. It also included questions on the need for additional knowledge or support.Part 3: Bowel Habits: this section investigated the frequency and characteristics of bowel movements, stool consistency, and constipation related symptoms. It used Likert scale questions (e.g., frequency rated from “Daily” (4) to “Never” (0)) to capture symptom severity and patterns.Part 4: Pharmacological and Non-pharmacological Interventions for Managing Constipation and Their Effectiveness: this section explored the strategies patients used to manage constipation. It includes pharmacological Interventions collects information on prescribed constipation medications, frequency of use, and perceived effectiveness It used Likert scale questions (rated from “Very effective” (3) to “Not effective at all (0)”. Also includes Non-Pharmacological Interventions collects information about lifestyle and dietary practices to alleviate constipation such as high-fiber diet, drinking water, and regular exercise, and other natural remedies.

To establish content validity, the tool was reviewed by a subject matter experts. Their feedback was used to refine the questionnaire’s language, ensure clarity, and confirm the relevance and coherence of each item. Additionally, a pilot study was conducted using a sample of 20 participants from the target population. This pilot testing helped assess the clarity, comprehensibility, and internal consistency of the items. Based on the pilot results, minor modifications were made to enhance the tool’s reliability and usability before proceeding with the main study (the pilot results were excluded from the research results).

### 2.4. Data Collection

Data was collected during personal visits (face-to-face interactions) to outpatient departments of hospital clinics in a private hospital, where participants were approached and screened for eligibility. The participants were approached by the researcher during their scheduled clinic visits and screened for eligibility based on the study’s inclusion and exclusion criteria. Once deemed eligible, participants were invited to complete a self-administered digital questionnaire using a Google Form, which was accessed via a QR code or iPad provided by the research team. The first section of the form contained detailed information about the study title, aim, importance, and informed consent. For those with literacy or vision difficulties, the researcher assisted by reading questions aloud and recording responses. Printed surveys were provided to participants who were unable to use the digital format.

### 2.5. Data Analysis

The data were analyzed using the Statistical Package for the Social Sciences (SPSS), version 23 (https://www.ibm.com/products/spss-statistics) (accessed on 3 January 2025) along with Jamovi (Version 2.7.6) software [46]. Descriptive statistics, such as percentages and frequencies, were computed for the demographic categorical variables, including age, sex, marital status, education level, occupation, symptoms and other variables. A nonparametric test was used to compare categorical variables. A *p*-value of less than 0.05 was considered indicative of statistical significance.

### 2.6. Ethical Consideration

The IRB was obtained from the college’s MACHS Institutional Review Board (IRB) committee (Reference number: SR/RP/201; date 30 December 2024). Furthermore, prior to the commencement of data collection, the approved IRB was formally shared with the administrative authorities of the participants hospital clinics. Based on this approval, the hospital administration granted permission for the research team to conduct the study within their premises. All data used in the study were obtained directly from the participants through a structured questionnaire. The first section of the questionnaire provided clear information regarding research title, aim, importance of research, confidentiality assurance, and the voluntary nature of participation. Participants were explicitly informed of their right to withdraw at any time without any consequences, and all responses remained anonymous and confidential.

## 3. Results

### 3.1. Demographic Characteristics of the Participants

A total of 382 individuals responded to the questionnaire; however, 82 responses were excluded from the analysis as those participants did not have a diagnosis of hypothyroidism. The use of a convenience sampling method and the inability to achieve a larger sample size were primarily due to practical and logistical constraints. Recruitment relied on voluntary participation through self-administered questionnaires, which inherently limits control over participant eligibility and response rates. Additionally, the strict inclusion criteria specifically requiring a confirmed diagnosis of hypothyroidism led to the exclusion of a substantial number of initial respondents who either did not meet this clinical criterion or provided incomplete data. These methodological choices were necessary to ensure the clinical relevance and data quality of the final sample, even though they may limit the representativeness of the broader hypothyroid population.

Table 1 summarizes the participants’ demographics and health-related characteristics, including age, gender, duration of hypothyroidism, and presence of chronic illnesses. Most respondents were between 41 and 60 years old (58.6%) and slightly more than half were female (51.7%). In terms of disease duration, 41.2% had been living with hypothyroidism for 1 to 5 years, while 39.1% reported having the condition for over 5 years. Among chronic conditions, cardiac disease was the most frequently reported (24%), followed by diabetes and other endocrine disorders, each affecting 11% of participants. Regarding the income levels employment status were not directly measures, as the structural healthcare framework in Saudi Arabia ensures equitable access to care for both Saudi nationals and legal residing expatriates. And most individuals who access private hospitals to receive care have health insurance covered their treatment. Thus, the influence of socioeconomic barriers such as inability to afford treatment is minimized in this context.

### 3.2. Participants’ Awareness of Bowel Complications Associated with Hypothyroidism

As presented in Table 2, the findings indicate that a substantial portion of participants lacked sufficient knowledge regarding the relationship between hypothyroidism and constipation. While 63% reported discussing their bowel habits with their physician, 37% had never initiated such discussions suggesting gaps in patient—provider of care communication regarding this critical symptom or the participants do not view constipation as a relevant symptom of their thyroid condition.

Furthermore, only 62.7% of participants were aware that hypothyroidism can contribute to constipation, leaving over one third (37.3%) unaware of this common and clinically significant complication. Given that constipation is among the most common gastrointestinal manifestations of hypothyroidism, the lack of awareness in more than one third of participants reflects a significant educational gap. This may contribute to underreporting of symptoms or mismanagement.

In terms of self-assessed knowledge, 42% described themselves as slightly knowledgeable, 33% as somewhat knowledgeable, and only 7.7% considered themselves very knowledgeable. These self-perceptions, coupled with high percentage of the participants (88.3%) expressed a need for more information or support. It provides an opportunity for improved patient education and clinical engagement in this area and to implement targeted interventions such as patient-centered counseling and educational leaflets.

Despite these gaps in knowledge, 42% of participants reported being somewhat satisfied with their current constipation management, while 41.3% remained neutral and 16.7% expressed dissatisfaction. These responses may reflect limited awareness of opti-mal treatment options or the symptoms due to a normalization of symptoms due to chronicity. The participants identified several factors that could increase the risk of their constipation symptoms, including coffee (59.7%), antihypertensive drugs (12%), and pain medication (10%).

### 3.3. Bowel Habit

Table 3 highlights the relationship between participants’ experiences of constipation and their hypothyroidism. The majority (97%) reported that their constipation became more severe after being diagnosed with hypothyroidism. Regarding constipation frequency, 34% experienced constipation twice per week, while 30.3% reported it occurring once weekly. Some factors were reported to worsen constipation symptoms, with coffee (59.7%), antihypertensive (12%), and pain medications (10%) being the most frequently mentioned. Many participants also reported additional symptoms such as anal fissures (18%) and abdominal pain (17.7%).

Figure 1 provides a clear picture of stool characteristics among participants, offering insight into both the frequency and type of bowel movements experienced. A substantial proportion of respondents reported separate hard lumps (62.3% frequently, 13% daily) and lumpy sausage-shaped stools (63.7% frequently, 12% daily), both of which are typical indicators of slow bowel transit. Similarly, sausage-shaped stools with cracks in the surface were also common, with 59.7% experiencing them frequently and 10% daily. By contrast, softer stool types were reported far less often. Only 4% had smooth, soft sausage or snake-like stools daily, and 23.3% frequently.

### 3.4. Pharmacological and Non-Pharmacological Interventions for Managing Constipation and Their Effectiveness

Figure 2 presents the frequency of different constipation management strategies used by patients with hypothyroidism. It shows that fiber supplements (58.3%), stool softeners (56.3%), and laxatives (57.7%) were among the most frequently used pharmacological treatments. For non-pharmacological approaches, increasing water intake (50.3%) and eating a fiber-rich diet (50.3%) were the most common choices. Notably, these two lifestyle measures were also the most consistently practiced daily, with 37% and 32.3% of participants, respectively, reporting regular use. Exercise was also a commonly reported strategy, with 46.3% of participants practicing it frequently. While probiotics and herbal or folk medicine were used less frequently daily, they still appear in the management routines of some participants (42.3% and 45.7% frequently, respectively).

Figure 3 shows how participants rated the effectiveness of different ways to manage constipation. Drinking more water was rated the most effective method, with 86.7% of participants reporting it as very effective. This was followed by eating fiber-diets (81.3%), using stool softeners (78.3%), and taking laxatives (72.3%), all of which had high effectiveness ratings and relatively low dissatisfaction rates (less than 10%) for all of them. Exercise and fiber supplements were also seen as helpful, with 73.6% and 68.3% rating them as very effective, though a few participants felt neutral or found them less useful. On the other hand, Probiotics and herbal or traditional remedies had more mixed feedback while more than half found them very effective (56.7% and 61.4%), they also had the highest number of people who felt they were less effective or not effective at all (21.7% and 18.3%).

### 3.5. Correlation Between Constipation Frequency and Other Factors

Table 4 presents the cross-tabulation reveals significant associations between constipation frequency and several factors, including age (*p* = 0.042), duration of hypothyroidism (*p* < 0.012), and constipation management methods such as laxatives (*p* = 0.022), stool softeners (*p* < 0.012), increased dietary fiber (*p* = 0.022), hydration (*p* = 0.042), exercise (*p* = 0.022), and herbal remedies (*p* = 0.042), with all *p*-value below 0.05. Patients aged 41–60 years and those with a longer history of hypothyroidism tended to report more frequent constipation. On the other hand, consistent use of stool softeners, higher fiber intake, and increased water consumption were associated with better bowel regularity. In contrast, no significant relationship was found between constipation frequency and gender (*p* = 0.522), fiber supplements (*p* = 0.222), or probiotics (*p* = 0.142). These results emphasize the importance of maintaining both lifestyle and pharmacological measures in managing constipation for people with hypothyroidism. Clinically, regular hydration, ad-equate fiber intake, and appropriate use of laxatives or stool softeners appear to play a key role in improving bowel function in hypothyroid patients.

Table 5 and Figure 4, summarizes the results of Spearman’s rho correlation analysis between three variables: the average constipation severity, constipation management approaches, and overall perceived effectiveness of constipation management. A moderate positive correlation was found between constipation severity and the use of management strategies (*r* = 0.431, *p* < 0.001), suggesting that individuals with more severe or frequent symptoms are more likely to adopt multiple approaches to manage their condition. Similarly, constipation severity showed a moderate positive correlation with treatment effectiveness (*r* = 0.503, *p* < 0.001), indicating that those with more severe constipation often reported better relief when appropriate interventions were used, possibly because effective strategies prompt greater symptom engagement and tracking. The strongest observed correlation was between constipation management and total effectiveness (*r* = 0.699, *p* < 0.001), indicating a strong positive relationship. This means that patients who actively use management strategies, whether pharmacological or non-pharmacological, tend to report higher effectiveness in relieving their symptoms. Furthermore, no significant correlation was found between disease duration and constipation characteristics (ρ = 0.021, *p* = 0.714), constipation management (ρ = 0.062, *p* = 0.285), or effectiveness of management (ρ = 0.066, *p* = 0.258).

## 4. Discussion

The study found that most participants were between 41 and 60 years (58.6%), with female making up a slightly larger proportion (51.7%). These findings are consistent with previous research which has identified both age and gender as important demographic factors influencing constipation, especially in people living with hypothyroidism. Ghanbari et al. (2024) reported similar findings, noting a higher prevalence of constipation in middle-aged females with hypothyroidism [1]. Other studies have established a strong correlation between female gender and constipation 0 [15,47], with some indicating that women may be twice as likely to experience it [12]. This gender disparity may stem from hormonal differences, slower colonic transit, or differing health-seeking behaviors. In Saudi Arabia, Alhusainy et al. (2018) also reported a higher prevalence of constipation among females, reinforcing the trend observed in Saudi populations [13]. Reported that constipation in hypothyroid patients was more common among older adults aged 60 and above, whereas our study focused on a slightly younger age group of 41–60 years. In contrast, a systematic review by Barberio et al. (2021) found no significant age-related differences in the prevalence of functional constipation, suggesting that age-related effects may be influenced more by comorbidities and lifestyle factors than age alone [12]. Our findings align with Samei et al. (2022) [48], who found a significant association between chronic constipation and hypothyroidism in pediatric patients. Yaylali et al. (2009) provided physiological evidence of delayed gastrointestinal motility in hypothyroid patients, providing a plausible biological explanation for the relationship [49].

In this study, 62.7% of participants were aware of the link between hypothyroidism and constipation, while 37.3% did not recognize this connection. This aligns with findings by Jadi and Moustaghit (2024) [50], who reported that although 88.7% of patients knew they had a hypo functioning thyroid gland, many lacked detailed knowledge about its symptoms and management. Similarly, Sethi, Khandelwal, and Vyas (2018) found that only 45.4% of patients correctly identified constipation as a symptom of hypothyroidism, highlighting the limited understanding of its clinical manifestations [51].

Furthermore, in the current study, 42% of participants described themselves as slightly knowledgeable and 33% as somewhat knowledgeable about hypothyroidism. The majority (88.3%) expressed a need for more information or support in managing their symptoms. These results align with Sethi et al. (2018) [51], who noted that 66.6% of patients had low knowledge levels, and only 12% demonstrated high awareness. Notably, 91.4% of their participants emphasized the importance of consulting a health care professional be-fore initiating treatment. Together, these findings highlight the urgent need to strengthen patient education and enhance physician–patient communication, particularly regarding the gastrointestinal effects of hypothyroidism.

The results show that 97% of participants experienced a worsening of constipation symptoms following their diagnosis of hypothyroidism, supporting earlier research that identifies constipation as a frequent complication in this condition, in particular, 34% reported having constipation twice a week, and 30.3% once a week, reflecting patterns typical of chronic constipation [52]. These findings are consistent with Ghanbari et al. (2024), who similarly observed that patients with hypothyroidism often experience reduced defecation frequency, consistent with slow transit constipation [1]

The study revealed that fiber supplements (58.3%), stool softeners (56.3%), and laxatives (57.7%) were the most frequently used pharmacological treatments for managing chronic constipation, aligning with previous research [53,54]. These treatments include bulking agents, stool softeners, and stimulant purgatives, with osmotic laxatives and stimulant laxatives recommended as first-line options when non-pharmacological measures fail [23,41,42,43]. In this study, stool softeners (78.3%) and laxatives (72.3%) were also reported as highly effective, consistent with evidence from systematic reviews and clinical guidelines issued by the American Gastroenterological Association and the American College of Gastroenterology [23]. Additionally, for cases of refractory constipation or irritable bowel syndrome with constipation (IBS-C), secretagogues such as lubiprostone, linaclotide, and plecanatide have demonstrated therapeutic benefits [40,41]. Further-more, the results also showed that while 61.4% of participants found herbal remedies to be effective, 8.3% reported them as ineffective, indicating variability in individual responses, a finding consistent with the findings of Dimidi et al. (2014) [55]. Similarly, Kanzaria et al. (2017) also noted that certain herbal treatments such as Vidanga have shown effectiveness for hypothyroidism-related constipation by improving intestinal motility [56,57].

The study found that the most common non-pharmacological strategies for managing chronic constipation were increasing water intake (50.3%), consuming a fiber-rich diet (50.3%), and engaging in regular exercise (46.3%). These approaches align with those that recommend lifestyle modifications as first-line interventions [53]. Participants also rated these methods highly effective, with 86.7% endorsing water consumption, 81.3% supporting a fiber-rich diet, and 73.6% recognizing exercise as beneficial. These perceptions are consistent with evidence linking hydration, dietary fiber, and physical activity to improvements in stool frequency and consistency [58,59,60,61]. However, while water consumption received the highest effectiveness rating from participants, some clinical reviews suggest that increased fluid intake may primarily benefit individuals who are de-hydrated and may not reliably improve bowel function in the general population [62,63,64]. Psyllium, a soluble fiber, can help increase stool bulk and frequency, particularly in individuals with low dietary fiber intake. However, it may cause side effects such as gas and bloating, and its impact on bowel transit time is limited [41,58,65]. Notably, fiber therapy may not benefit patients with slow-transit constipation or dyssynergic defecation may experience little to no benefit [58,66]. Physical activity has also been linked to constipation relief in several observational studies [67,68], although findings remain inconsistent. While regular exercise is generally associated with improved bowel movements and enhanced quality of life, its effect on constipation severity appears to vary across studies [69].

The current study found significant associations between constipation frequency and several factors such as age, duration of hypothyroidism, and various management strategies, including laxatives, stool softeners, dietary fiber, hydration, physical activity, and herbal remedies. These findings are consistent with previous research indicating that advancing aging and longer hypothyroidism duration increase the risk of constipation, and that effective management often requires a combination of lifestyle changes and medical interventions [5,53]. Additionally, Shen et al. (2019) and Eswaran et al. (2013) highlighted the contribution of low fiber intake and inadequate hydration to chronic constipation [58,62,63], while, research by Krogh, Chiarioni, and Whitehead (2017) supports the effectiveness of laxatives, stool softeners, and exercise in relieving symptoms of chronic or functional constipation [40]. Additionally, exercise has been shown to enhance intestinal gas clearance and increase bowel movement frequency [60,61].

Furthermore, the study results found that frequent use of stool softeners, high dietary fiber intake, and increased water consumption significantly improved bowel habits. These results align with the American Gastroenterological Association’s recommendations for managing chronic constipation using a combination of pharmacological and non-pharmacological interventions [5]. In contrast, factors such as gender, fiber supplements, and probiotics showed no significant relationship with constipation frequency. This differs from previous research, Barberio et al. (2021) reported a higher prevalence of constipation in women, and Dimidi et al. (2014) found that probiotic strains could improve stool frequency and consistency [12,55]. These discrepancies may be due to variations in study populations, the specific probiotic strains examined, or differences in participants’ adherence to fiber supplementation regimens. Correlation analysis further revealed a moderate positive relationship between constipation characteristics and both management strategies and overall treatment effectiveness. The strongest correlation was observed between constipation management and perceived effectiveness, reinforcing the importance of tailored, consistent management approaches. These findings are supported by Rao and Brenner (2021) [42,43], who emphasized that evidence-based management strategies can significantly improve constipation and quality of life. Our findings emphasize the importance of rehabilitation-oriented nursing care in managing hypothyroidism-related constipation. Nurses play a key role in promoting self-management strategies such as adequate hydration, fiber intake, and physical activity, which align with rehabilitation goals of patient autonomy and symptom control. Evidence shows that nurse-led self-management interventions can reduce constipation severity and improve treatment adherence [70]. Similarly, implementing structured bowel care protocols in rehabilitation settings has improved care practices and documentation, supporting better long-term management [71]. These insights highlight the value of integrating constipation management into broader rehabilitation and chronic care programs, ultimately enhancing patients’ quality of life.

### 4.1. Implications and Recommendations

Based on the findings, healthcare providers, especially nurses, should develop and integrate education programs about bowel health into the routine care plans of individuals with hypothyroidism. The education program should consider the gap in patient knowledge regarding the relationship between hypothyroidism and constipation also encouraging lifestyle practices such as adequate water intake, fiber diet, and regular physical activity can improve constipation out-comes. Structured, nurse-led programs could further enhance patient knowledge and promote effective self-management. The development and implementation of such programs could play a vital role in improving patients’ quality of life by enhancing their ability to recognize, report, and manage symptoms more effectively. Moreover, patient-centered education would not only empower individuals in their self-care but also contribute to earlier diagnosis and timely treatment, thereby reducing the burden of chronic complications.

Future research should examine the long-term effects of combining pharmacological and non-pharmacological strategies for managing constipation among hypothyroid patients. Comparative studies are needed to assess different patient education methods and their impact on symptom control would be valuable.

Studies involving larger, clinically verified samples with access to medical records and thyroid profiles, and using advanced statistical models to control confounders, are recommended to clarify the relationship between hypothyroidism severity and constipation. Including diverse populations across Saudi Arabia and accounting for other demographic factors, socioeconomic status, and income would further enrich interpretation and enhance the generalizability of findings.

### 4.2. Limitation

This study has several limitations. First, its cross-sectional design makes it difficult to establish cause-and-effect relationships between hypothyroidism, constipation, and treatment effectiveness. Second, reliance on self-reported data may introduce recall and reporting bias, which could affect the accuracy of the results. Third, one notable limitation is relatively limited sample size and use of convenience sampling. This reduction in sample size may limit the generalizability of the findings and reduce the statistical power to detect subtle associations or subgroup differences. In addition to using convenience sampling may have introduced selection bias, limiting the representativeness of the sample and, therefore, the generalizability of the results. Finally, the lack of objective clinical data, like thyroid hormone levels or colon transit times, limits the depth of clinical interpretation. Another important limitation, the absence of objective clinical data, such as thyroid hormone levels or colon transit assessments, limited the depth of clinical interpretation in this study. Finally, the inability to control for confounding factors, including gastrointestinal conditions, medications use, and psychosocial influences, all of which may influence constipation outcomes and thereby affect the reliability of the findings. Future research with larger, clinically verified samples and detailed medical data is needed to better account for these variables and strengthen causal interpretations.

## 5. Conclusions

This study highlights the prevalence of constipation among individuals with hypothyroidism and identifies significant associations between constipation frequency and factors such as age, disease duration, lifestyle or treatment interventions. Both pharmacological interventions, such as stool softeners, laxatives, and fiber supplements and non-pharmacological measures, such as hydration, dietary fiber, and regular exercise, were commonly used and perceived as effective, especially increasing both water intake and fiber-rich diet. A strong correlation was found between consistent constipation management and perceived treatment effectiveness. However, some strategies, including fiber supplements and probiotics, did not show a statistically significant association with constipation frequency, suggesting differences in how individuals respond. Overall, these findings emphasize the importance of adopting personalized, multi-modal management plans that align with each patient’s needs, health profile, and preferences.

## Figures and Tables

**Figure 1 nursrep-15-00354-f001:**
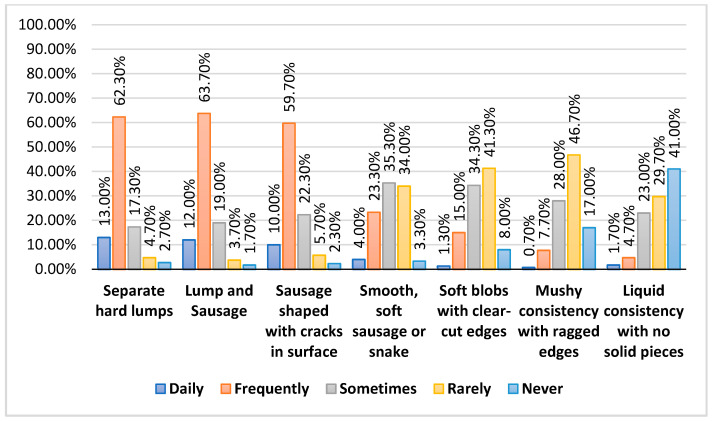
Stool Characteristics Among Participants.

**Figure 2 nursrep-15-00354-f002:**
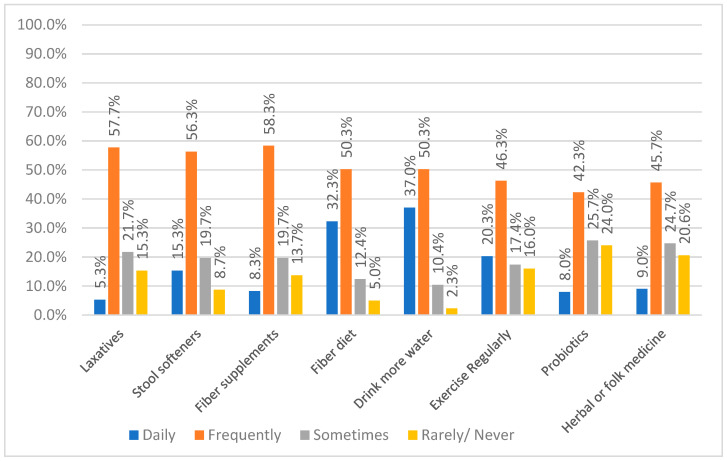
Frequency of Constipation Management Methods Used by Participants.

**Figure 3 nursrep-15-00354-f003:**
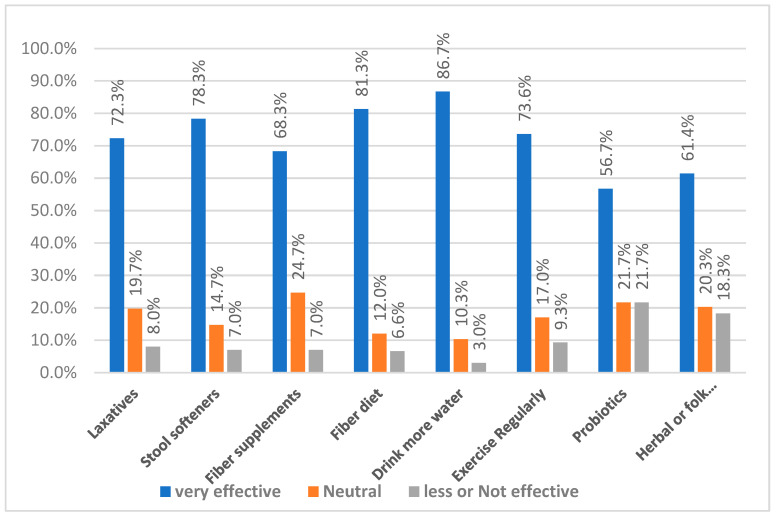
Perceived Effectiveness of Constipation Management Methods.

**Figure 4 nursrep-15-00354-f004:**
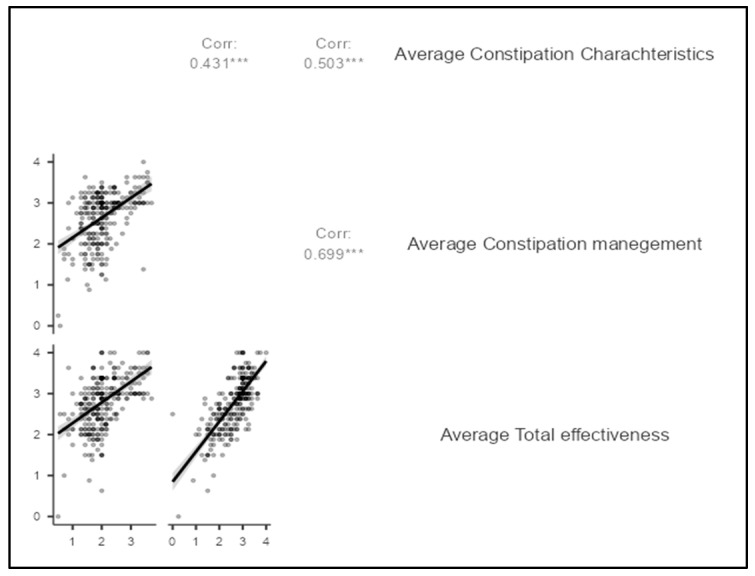
Scatter plot for Spearman’s Correlation Between Constipation Severity, Management Strategies, and Perceived Effectiveness. *** (three stars) = *p* < 0.001 (highly significant at the 0.1% level).

**Table 1 nursrep-15-00354-t001:** Demographic and Health Characteristics of Participants (*N* = 300).

Variables	Category	Frequency (*n*)	%
**Age**	18–30	31	10.3%
	31–40	59	19.7%
	41–50	106	35.3%
	51–60	70	23.3%
	61–70	23	7.7%
	71 Above	11	3.7%
**Gender**	Female	155	51.7%
	Male	145	48.3%
**Duration of Hypothyroidism**	Less than 1 year	59	19.7%
	1–5 years	124	41.2%
	More than 5 years	117	39.1%
	No chronic disease	94	31.3%
**Chronic Disease**	Diabetes	33	11.0%
	Other Endocrine disease	33	11.0%
	Cardiac disease	72	24.0%
	Respiratory Disorders	30	10.0%
	Muscular chronic disorders	26	8.7%
	Neuro chronic disorders	12	4.0%

**Table 2 nursrep-15-00354-t002:** Participants’ Knowledge of Hypothyroidism and Constipation (*N* = 300).

Item	Response	Frequency (*n*)	Percentage (%)
Did you discuss your bowel status with your doctor?	Yes	189	63.0%
	No	111	37.0%
Do you know that Hypothyroidism causes constipation?	Yes	188	62.7%
	No	112	37.3%
Participants evaluate their knowledge about the effect of hypothyroidism on their bowel status	Not knowledgeable at all	52	17.3%
	Slightly knowledgeable	126	42.0%
	Somewhat knowledgeable	99	33.0%
	Very knowledgeable	23	7.7%
Need more information or support on managing constipation	Yes	265	88.3%
	Not Sure	29	9.7%
	No	6	2.0%
Satisfaction with current constipation management	Very satisfied	4	1.3%
	Somewhat satisfied	122	40.7%
	Neutral	124	41.3%
	Dissatisfied	42	14.0%
	Very dissatisfied	8	2.7%
Medication will increase your risk of constipation	None	23	7.7%
	Coffee	179	59.7%
	Calcium	20	6.7%
	Iron	12	4.0%
	Antihypertensive drugs	36	12.0%
	Pain medication (Opioid as morphine, codeine, etc.)	30	10.0%

**Table 3 nursrep-15-00354-t003:** Constipation Characteristics Among Participants (*N* = 300).

Variable	Category	Frequency (*n*)	Percentage (%)
Has your bowel changed after the diagnosis of hypothyroidism?	No changes	8	2.7%
	Yes, increased constipation	291	97.0%
	Yes, other changes	1	0.3%
What is the constipation frequency?	Once daily	75	25.0%
	More than once daily	30	10.0%
	Once per week	91	30.3%
	twice per week	102	34.0%
	More than a week	2	0.7%
History of other symptoms with constipation	Dark color	27	9.0%
	Foul odor	35	11.7%
	Anal fissure	54	18.0%
	Hemorrhoids	16	5.3%
	Abdominal discomfort/bloating	31	10.3%
	Fecal impaction	41	13.7%
	Abdominal pain	53	17.7%
	Feeling of incomplete evacuation	26	8.7%
	Straining during bowel movements	3	1.0%
	Hard or lumpy stools	6	2.0%
	None	8	2.6%

**Table 4 nursrep-15-00354-t004:** Association Between Constipation Frequency and Patient Characteristics, Lifestyle Modifications, and Management Strategies (*N* = 299).

Variable	Frequency (*n*)	Once Daily	More Than Once Daily	Once Per Week	Twice Per Week	More Than a Week	Test Statistic
		(**N = 75**)	(**N = 30**)	(**N = 90**)	(**N = 102**)	(**N = 2**)	
**Age (years)**	299						**Χ220 = 32.54, P = 0.04^2^**
18–30		0.111	0.14	0.02	0.114	0.00	
31–40		0.214	0.26	0.217	0.222	0.00	
41–50		0.323	0.411	0.544	0.3 28	0.00	
51–60		0.319	0.14	0.220	0.2 25	1.02	
61–70		0.0 3	0.25	0.15	0.1 9	0.00	
71 Above		0.1 5	0.00	0.02	0.0 4	0.00	
**Gender: Male**	299	0.539	0.310	0.544	0.5	0.51	**Χ24 = 3.23, P = 0.52^2^**
**Duration of hypothyroidism**	299						**Χ28 = 28.22, P < 0.01^2^**
Less than 1 year		0.324	0.39	0.19	0.217	0.00	
1–5 Years		0.326	0.310	0.331	0.5 55	0.51	
More than 5 years		0.325	0.411	0.650	0.330	0.51	
**Use Laxatives**	299						**Χ212 = 23.76, P = 0.02^2^**
Daily		0.1 6	0.1 2	0.03	0.0 5	0.00	
Frequently		0.541	0.823	0.761	0.5 46	0.51	
Sometimes		0.215	0.14	0.112	0.3 34	0.00	
Rarely		0.213	0.01	0.214	0.2 17	0.51	
**Use Stool softeners**	299						**Χ212 = 33.22, P < 0.01^2^**
Daily		0.212	0.12	0.1	0.225	0.00	
Frequently		0.541	0.823	0.760	0.444	0.51	
Sometimes		0.319	0.14	0.111	0.2 25	0.00	
Rarely		0.03	0.01	0.112	0.18	0.51	
**Use Fiber supplements**	299						**Χ212 = 15.31, P = 0.22^2^**
Daily		0.14	0.13	0.17	0.1 11	0.00	
Frequently		0.644	0.824	0.655	0.551	0.51	
Sometimes		0.218	0.12	0.215	0.224	0.00	
Rarely		0.19	0.01	0.113	0.216	0.51	
**Increase dietary fiber**	299						**Χ212 = 23.97, P = 0.02^2^**
Daily		0.433	0.26	0.220	0.4 37	0.51	
Frequently		0.429	0.721	0.654	0.5 46	0.51	
Sometimes		0.16	0.01	0.214	0.1 15	0.00	
Rarely		0.17	0.12	0.02	0.04	0.00	
**Drink more water**	299						**Χ212 = 22.10, P = 0.04^2^**
Daily		0.535	0.3 8	0.325	0.4 42	0.51	
Frequently		0.428	0.720	0.657	0.4 44	0.51	
Sometimes		0.18	0.01	0.17	0.115	0.00	
Rarely		0.14	0.01	0.01	0.01	0.00	
**Exercise Regularly**	299						**Χ212 = 24.58, P = 0.02^2^**
Daily		0.321	0.25	0.113	0.2 21	0.51	
Frequently		0.324	0.721	0.652	0.4 42	0.00	
Sometimes		0.214	0.01	0.214	0.2 21	0.51	
Rarely		0.216	0.13	0.111	0.2 18	0.00	
**Take probiotics**	299						**Χ212 = 17.40, P = 0.14^2^**
Daily		0.17	0.13	0.03	0.1 11	0.00	
Frequently		0.427	0.412	0.546	0.4 41	0.00	
Sometimes		0.323	0.411	0.218	0.2 25	0.00	
Rarely		0.218	0.14	0.323	0.225	1.02	
**Use herbal/folk medicine**	299						**Χ212 = 21.59, P = 0.04^2^**
Daily		0.19	0.13	0.0 4	0.1 10	0.51	
Frequently		0.429	0.619	0.650	0.4 38	0.00	
Sometimes		0.217	0.25	0.220	0.3 32	0.00	
Rarely		0.320	0.13	0.216	0.2 22	0.51	
N is the number of non-missing value.							

**Table 5 nursrep-15-00354-t005:** Spearman’s Correlation Between Disease Duration, Constipation Severity, Management Strategies, and Perceived Effectiveness (*N* = 299).

		Disease Duration	Constipation Characteristics	Constipation Management	Effectiveness of Management
**Disease Duration**	Spearman’s rho	—			
	df	—			
	*p*-value	—			
**Constipation Characteristics**	Spearman’s rho	0.021	—		
	df	297	—		
	*p*-value	0.714	—		
**Constipation Management**	Spearman’s rho	0.062	0.431	—	
	df	297	298	—	
	*p*-value	0.285	<0.001	—	
**Effectiveness of Management**	Spearman’s rho	0.066	0.503	0.699	—
	df	297	298	298	—
	*p*-value	0.258	<0.001	<0.001	—

## Data Availability

The datasets used and/or analyzed during the current study are available from the first author on reasonable request.

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
