# Peer review of "Integrated Management of Constipation in Hypothyroidism: Evaluating Pharmacological and Non-Pharmacological Interventions"

_nursrep, 2025, doi:10.3390/nursrep15100354_

Round 1
Reviewer 1 Report
Comments and Suggestions for Authors
Thank you for inviting me to review the article “Integrated Management of Constipation in Hypothyroidism: Evaluating Pharmacological and Non-Pharmacological Interventions.” The article addresses a clinical issue of great practical importance, namely constipation in patients with hypothyroidism. The authors rightly point to the key role of thyroid hormones in regulating metabolic processes, including intestinal peristalsis. They also emphasize that patients' awareness of the relationship between the disease and gastrointestinal symptoms varies—some patients do not recognize this relationship, while others are able to identify both the mechanisms and ways of dealing with the problem. The article also highlights the impact of chronic constipation on quality of life and mental well-being, which is an important aspect of an interdisciplinary approach to the patient. The strength of the article lies in its apt choice of topic, which combines medical knowledge with the patients' perspective. The authors rightly emphasize both the biological basis of the phenomenon (thyroid hormone deficiency) and environmental and behavioral factors (diet, physical activity). The article has an educational dimension, as it makes patients aware of the importance of symptoms and the need for a comprehensive therapeutic approach. key data:
- Characteristics of the problem: Chronic constipation is a complex clinical condition influenced by demographic, social, and economic factors. The study lacks in-depth data on which elements, which impoverishes the interpretative context.
- Research sample; The methodology section indicates that the data was obtained from hospitals, but no details are provided on how it was obtained.
- The recruitment of participants is described as having been conducted during personal meetings at the clinic, which contradicts the earlier information about hospitals. It is necessary to clearly specify the source of the data and the recruitment procedure.
- Research methodology; The survey was addressed to patients, but the authors do not indicate whether the relevant institutions gave their consent to conduct the study or who made the patient data available and on what basis. The lack of this information raises ethical and formal concerns.
- It is also necessary to provide details on the validation of the research tool and the possible approval of the study by a bioethics committee.
- Empirical part; Despite the use of multidimensional models, the statistical analysis does not take into account significant confounding variables (e.g., coexisting gastrointestinal diseases, medications taken, psychosocial factors). The lack of control of these variables limits the reliability of the conclusions.
- The data presented in the table are aggregated in a way that makes them difficult to interpret; it is recommended that the results be presented in a more transparent and detailed form, e.g., broken down into patient subgroups.
- The text also does not present detailed statistical analyses. The results are limited to descriptions, without significance tests or analysis of the relationship between the degree of hypothyroidism and the severity of constipation. As a result, the conclusions are observational and descriptive rather than evidence-based.
- The section on participants' knowledge of the relationship between hypothyroidism and constipation needs improvement. The result itself is significant and highlights the need for patient education, but its interpretation is limited by methodological shortcomings.
- Despite the use of multidimensional models, the statistical analysis does not take into account potential confounding variables (including coexisting gastrointestinal diseases, medications used, or psychosocial factors). In addition, the presentation of data in tables is too condensed, which makes it difficult to evaluate detailed results and limits their transparency.
- At the same time, despite the limitations indicated, the results of the study draw attention to the very low level of patient knowledge about the relationship between thyroid disease and constipation. This fact should be an impetus for the development and implementation of educational programs in primary care facilities and hospitals, which may improve the quality of life of patients and facilitate early diagnosis and treatment of symptoms.
- Summary; The article addresses an important and publishable topic, but its methodological and empirical parts require significant additions and clarification. In its current form, the paper does not fully meet the standards of scientific research and should be revised before being accepted for publication.
Author Response
Dear Reviewer,
We are grateful for the opportunity to revise our manuscript entitled “Integrated Management of Constipation in Hypothyroidism: Evaluating Pharmacological and Non-Pharmacological Interventions” (Manuscript ID: nursrep-3849959). We sincerely thank you and the reviewers for your thoughtful feedback, which has been invaluable in strengthening our work. Below, we provide a point-by-point response outlining how we have addressed each comment and the corresponding revisions made in the manuscript.
Comments and Suggestions for Authors.
Thank you for inviting me to review the article “Integrated Management of Constipation in Hypothyroidism: Evaluating Pharmacological and Non-Pharmacological Interventions.” The article addresses a clinical issue of great practical importance, namely constipation in patients with hypothyroidism. The authors rightly point to the key role of thyroid hormones in regulating metabolic processes, including intestinal peristalsis. They also emphasize that patients' awareness of the relationship between the disease and gastrointestinal symptoms varies—some patients do not recognize this relationship, while others are able to identify both the mechanisms and ways of dealing with the problem. The article also highlights the impact of chronic constipation on quality of life and mental well-being, which is an important aspect of an interdisciplinary approach to the patient. The strength of the article lies in its apt choice of topic, which combines medical knowledge with the patients' perspective. The authors rightly emphasize both the biological basis of the phenomenon (thyroid hormone deficiency) and environmental and behavioral factors (diet, physical activity). The article has an educational dimension, as it makes patients aware of the importance of symptoms and the need for a comprehensive therapeutic approach. key data:
Comment: Characteristics of the problem: Chronic constipation is a complex clinical condition influenced by demographic, social, and economic factors. The study lacks in-depth data on which elements, which impoverishes the interpretative context.
Response: We appreciate the reviewer’s valuable observation regarding the multifactorial nature of chronic constipation, particularly its association with demographic, social, and economic factors. In our study, several of these variables were addressed: demographic data included age, sex, and disease duration, with the exception of educational level; and social factors such as lifestyle, physical activity, dietary habits, and hydration were also incorporated. However, variables such as social support and psychosocial stressors, as well as economic aspects including income level, insurance coverage, and access to healthcare, were not examined.
Regarding the income levels were not directly measures, as the structural healthcare framework in Saudi Arabia ensures equitable free access to care for Saudi nationals residing expatriates. And most of individuals (Saudi and non-Saudi ) who access private hospitals to receive care they have health insurance covered their treatment. Thus, the influence of socioeconomic barriers such as inability to afford treatment is minimized in this context. . (a paragraph added in results section : 269-273).
We fully we acknowledge the merit of the reviewer’s suggestion and agree that future studies would benefit from integrating a broader set of socio-economic indicators, such as educational level, occupation, and household income, to enrich the interpretive context and strengthen the generalizability of findings. However, in the present study, we deliberately opted to keep the questionnaire concise, as excessive length has been shown to negatively impact participant completion rates and data quality. To maintain participant engagement and minimize survey fatigue, we prioritized questions directly aligned with the study’s core objectives (added a paragraph in 4.1. recommendation to explain that 532-537).
Comment: Research sample; The methodology section indicates that the data was obtained from hospitals, but no details are provided on how it was obtained.
Response: A non-probability convenience sample was used to collect information from patients with hypothyroidism from outpatient clinics. The recruitment specifically conducted during fac-to face encounters in outpatient endocrinology and internal medicine clinics within these hospitals. (added a paragraph in 2.2. Sample section to explain that 145-147).
Comment: The recruitment of participants is described as having been conducted during personal meetings at the clinic, which contradicts the earlier information about hospitals. It is necessary to clearly specify the source of the data and the recruitment procedure.
Response: Data were collected during personal visits (face-to-face interactions) to outpatient departments of hospital clinics in a private hospital, where potential participants were approached and screened for eligibility. The participants were approached by the researcher during their scheduled clinic visits and screened for eligibility based on the study’s inclusion and exclusion criteria. (added a paragraph in 2.2. Sample to explain that 145-147).
Once deemed eligible, participants were invited to complete a self-administered digital questionnaire using a Google Form, which was accessed via a QR code or iPad provided by the research team. The first section of the form contained detailed information about the study title, aim, importance, and informed consent. (added a paragraph in 2.4. Data collection 218-225).
Comment: Research methodology; The survey was addressed to patients, but the authors do not indicate whether the relevant institutions gave their consent to conduct the study or who made the patient data available and on what basis. The lack of this information raises ethical and formal concerns.
Response: We sincerely appreciate the reviewer’s attention to ethical considerations, and we would like to clarify the measures taken to ensure that all institutional and procedural requirements were appropriately followed.
Ethical approval for the study was obtained from the Institutional Review Board (IRB) of the College of Medical Applied Sciences at MACHS (Reference number: SR/RP/201; date: 30/12/2024). Furthermore, prior to the commencement of data collection, the approved IRB documentation was formally shared with the administrative authorities of the participating hospital clinics. Based on this approval, the hospital administration granted permission for the research team to engage with potential participants during their clinic visits and conduct the study within their premises. 2.6 Data ethical consideration 238-247).Importantly, all data used in the study were obtained directly from participants through a structured questionnaire, and no clinical or personal data were extracted from hospital records or accessed through medical databases. This ensured full compliance with data protection protocols and patient confidentiality standards. The informed consent process was integral to the study: the first section of the questionnaire provided clear information regarding the research title, objectives, confidentiality assurances, and the voluntary nature of participation. Participants were explicitly informed of their right to withdraw at any time without any consequences, and all responses remained anonymous and confidential. added a paragraph in data collection section 218-225, and 2.6. ethical consideration 238-247).
Data were obtained directly from patients via questionnaires, and no access to hospital records was sought; therefore, detailed information on comorbid gastrointestinal conditions or complete medication histories could not be comprehensively included in the analysis.
Comment: It is also necessary to provide details on the validation of the research tool and the possible approval of the study by a bioethics committee.
Response: The questionnaire used in this study was developed by the researcher following an extensive review of the relevant literature to ensure alignment with the study objectives and existing knowledge on hypothyroidism and chronic constipation. To establish content validity, the tool was reviewed by a subject -matters experts. Their feedback was used to refine the questionnaire’s language, ensure clarity, and confirm the relevance and coherence of each item. Additionally, a pilot study was conducted using sample of 20 participants from the target population. This pilot testing helped assess the clarity, comprehensibility, and internal consistency of the items. Based on the pilot results, minor modifications were made to enhance the tool’s reliability and usability before proceeding with the main study (the pilot results were excluded from the research results). (added paragraph in 2.3. Study Tools section 189- 198)..
With respect to ethical approval, the research was reviewed and approved by the Institutional Review Board (IRB) of the College of Medical Applied Sciences at MACHS (Reference number: SR/RP/201; date: 30/12/2024). The IRB’s review ensured that all ethical considerations, including informed consent, participant rights, and data confidentiality, were addressed in compliance with institutional and international ethical research standards. (ethical consideration section 238-247).
Comment: Empirical part; Despite the use of multidimensional models, the statistical analysis does not take into account significant confounding variables (e.g., coexisting gastrointestinal diseases, medications taken, psychosocial factors). The lack of control of these variables limits the reliability of the conclusions.
Response: Thank you for your valuable feedback. We appreciate the opportunity to clarify the methodological rigor applied in our study. While it is true that statistical models may be limited in capturing all potential confounding variables, considerable efforts were made during the study design to mitigate such biases at the recruitment stage. Specifically, the study excluded those individuals with coexisting gastrointestinal conditions (e.g., irritable bowel syndrome, inflammatory bowel disease, colorectal cancer), recent abdominal or pelvic surgery, and those on medications known to significantly impact bowel motility (such as opioids, anticholinergics, iron supplements, and certain antidepressants) were systematically excluded. Furthermore, participants with severe psychiatric illness or cognitive impairment were also not included, ensuring reliable self-reported data. These rigorous exclusion criteria were intentionally implemented to minimize the influence of non-thyroidal causes of constipation and other confounders, thereby enhancing the internal validity of our findings. (this is explained in sample section exclusion criteria 164-172)
Nonetheless, we acknowledge the inherent limitations of observational research and agree that future studies may benefit from incorporating additional statistical controls for psychosocial and residual confounding factors. (added in limitation section 551-556
We agree with the reviewer that these confounders may influence both the presence and severity of constipation and, therefore, represent an important factor in interpreting the findings. To address this concern, we have revised the limitations section of the manuscript to explicitly acknowledge that the absence of control for such variables may limit the generalizability and reliability of the conclusions.
Future research with larger, clinically verified samples and access to medical records, detailed medication profiles, and psychosocial assessments would allow for more robust statistical modeling and adjustment for confounding factors. This would strengthen the causal interpretation of the relationship between hypothyroidism, constipation, and management outcomes. ((added paragraph in 4.2. Limitation 542-556).
Comment: The data presented in the table are aggregated in a way that makes them difficult to interpret; it is recommended that the results be presented in a more transparent and detailed form, e.g., broken down into patient subgroups.
Response: We appreciate the reviewer’s suggestion regarding the presentation of Table 3. We divided the table data to one small table (table 3) and one figure (figure 1) to be simple presentation. . ( added in data analysis section 316- 324)
Comment: The text also does not present detailed statistical analyses. The results are limited to descriptions, without significance tests or analysis of the relationship between the degree of hypothyroidism and the severity of constipation. As a result, the conclusions are observational and descriptive rather than evidence based.
Response: We sincerely appreciate the reviewer’s observation regarding the depth of statistical analysis.
We added a correlation between disease duration and constipation characteristics (severity), constipation management , and effectiveness of management. There is no significant correlation was found . ( table 5 and 384-387)
The primary aim of this study was to explore hypothyroidism patients’ knowledge and practices regarding constipation and to evaluate the effectiveness of pharmacological and non-pharmacological management approaches. In line with this aim, and the research design which is cross sectional descriptive study, the analysis included both descriptive and inferential statistics.
While descriptive analyses were used to present participants’ demographic and health characteristics, levels of awareness, stool patterns, and reported management strategies, we also conducted correlation and association tests to move beyond purely descriptive observations. Specifically, the results include:
- Association analyses between constipation frequency and patient characteristics, lifestyle modifications, and management strategies.
- Spearman’s correlation analyses examining the relationship between constipation severity, management strategies, and perceived effectiveness.
These analyses were performed to capture patterns of association between variables, thereby strengthening the evidence base of our findings and aligning them with the exploratory nature of the study. However, we acknowledge that the research did not include tests specifically linking the degree of hypothyroidism with the severity of constipation, as this was beyond the intended scope of the study and would require access to detailed biochemical thyroid profiles, which were not collected from medical records.
We agree with the reviewer that further research incorporating such clinical measures, alongside more advanced statistical modeling, would provide a deeper understanding of the causal relationship between hypothyroidism severity and constipation outcomes. We have now emphasized this as a limitation in the discussion and suggested it as an important direction for future studies.
added a paragraph in 4.1. recommendation to explain that 514-525 ).
Comment: The section on participants' knowledge of the relationship between hypothyroidism and constipation needs improvement. The result itself is significant and highlights the need for patient education, but its interpretation is limited by methodological shortcomings.
Response: More detail was added to interpret the section 3.2 on participants' awareness in data analysis section (added paragraph in data analysis section 3.2 participants awareness of bowel constipation 278-302)
As presented in Table 2, the findings indicate that a substantial portion of participants lacked sufficient knowledge regarding the relationship between hypothyroidism and constipation. While 63% reported discussing their bowel habits with their physician, 37% had never initiated such discussions suggesting potential gaps in patient – provider of care communication regarding this critical symptom or the participants do not view constipation as a relevant symptom of their thyroid condition. Furthermore, only 62.7% of participants were aware that hypothyroidism can contribute to constipation, leaving over one third (37.3%) unaware of this common and clinically significant complication. Given that constipation is among the most common gastrointestinal manifestations of hypothyroidism, the lack of awareness in more than one third of participants reflects a significant educational gab. This may contribute to underreporting of symptoms or mismanagement. In terms of self-assessed knowledge, 42% described themselves as slightly knowledgeable, 33% as somewhat knowledgeable, and only 7.7% considered themselves very knowledgeable. These self-perceptions, coupled with high percentage of the participants (88.3%) expressed a need for more information or support. It provides an opportunity for improved patient education and clinical engagement in this area and implement targeted interventions such as patient centered counseling, educational leaflets. Despite these gabs in knowledge, 42% of participants reported being somewhat satisfied with their current constipation management, while 41.3% remained neutral and 16.7% expressed dissatisfaction. These responses may reflect limited awareness of optimal treatment options or the symptoms due to a normalization of symptoms due to chronicity.
Comment: Despite the use of multidimensional models, the statistical analysis does not take into account potential confounding variables (including coexisting gastrointestinal diseases, medications used, or psychosocial factors). In addition, the presentation of data in tables is too condensed, which makes it difficult to evaluate detailed results and limits their transparency.
This comment is repeated , we explained it in the previous point
Comments: At the same time, despite the limitations indicated, the results of the study draw attention to the very low level of patient knowledge about the relationship between thyroid disease and constipation. This fact should be an impetus for the development and implementation of educational programs in primary care facilities and hospitals, which may improve the quality of life of patients and facilitate early diagnosis and treatment of symptoms.
Response: We sincerely thank the reviewer for this valuable observation. We fully agree that our findings highlight a considerable gap in patient knowledge regarding the relationship between hypothyroidism and constipation.
(Added paragraph in 4.1. implication and recommendation 532-537)
Based on the findings, healthcare providers, especially nurses, should develop and integrate education programs about bowel health into the routine care plans of individuals with hypothyroidism. The education program should consider the gap in patient knowledge regarding the relationship between hypothyroidism and constipation also encouraging lifestyle practices such as adequate water intake, fiber diet, and regular physical activity can improve constipation out-comes. Structured, nurse-led programs could further enhance patient knowledge and promote effective self-management. The development and implementation of such programs could play a vital role in improving patients’ quality of life by enhancing their ability to recognize, report, and manage symptoms more effectively. Moreover, patient-centered education would not only empower individuals in their self-care but also contribute to earlier diagnosis and timely treatment, thereby potentially reducing the burden of chronic complications.
Summary; The article addresses an important and publishable topic, but its methodological and empirical parts require significant additions and clarification. In its current form, the paper does not fully meet the standards of scientific research and should be revised before being accepted for publication.
General Improvements
In addition to addressing the specific comments, we carefully reviewed the manuscript for clarity, coherence, and formatting. Minor language edits have been made to improve readability.
We believe these revisions have substantially improved the manuscript and aligned it more closely with the aims of the Special Issue. We are grateful for the reviewers’ and editors’ insightful comments, which have guided us in enhancing the scientific rigor and practical contribution of this study.
Thank you for considering our revised submission. We hope the revised manuscript will now be suitable for publication.
With kind regards,
On behalf of all authors,

Reviewer 2 Report
Comments and Suggestions for Authors
The manuscript entitled “Integrated Management of Constipation in Hypothyroidism: Evaluating Pharmacological and Non-Pharmacological Interventions” recommends health education and lifestyle guidance in care plans to enhance patient outcomes and self-management.
Why the subheading "Aim" before "Methods"?
To explore hypothyroidism patients’ knowledge and practice regarding constipation 135?
Elaborate more about the exclusion and inclusion criteria of this study.
Make a section-wise detail of study tools??
Table 3 should be simpler in presentation??
Figure 1. Frequency of Constipation Management Methods Used by Participants. 264
Make the Y-axis up to 100%.
Figure 2. Perceived Effectiveness of Constipation Management Methods. 277
Please represent it in another way?
Table 5. Spearman’s Correlation Between Constipation Severity, Management Strategies, and Per-309?
Could you please try to create a graphical presentation of this data?
Why does the discussion have subheadings??
Author Response
Dear Reviewer,
Dear Editor,
We are grateful for the opportunity to revise our manuscript entitled “Integrated Management of Constipation in Hypothyroidism: Evaluating Pharmacological and Non-Pharmacological Interventions” (Manuscript ID: nursrep-3849959). We sincerely thank you and the reviewers for your thoughtful feedback, which has been invaluable in strengthening our work. Below, we provide a point-by-point response outlining how we have addressed each comment, and the corresponding revisions made in the manuscript.
The manuscript entitled “Integrated Management of Constipation in Hypothyroidism: Evaluating Pharmacological and Non-Pharmacological Interventions” recommends health education and lifestyle guidance in care plans to enhance patient outcomes and self-management.
Comments:
- Why the subheading "Aim" before "Methods"?
- To explore hypothyroidism patients’ knowledge and practice regarding constipation 135?
Response: As suggested, the subheading “Aim” has been removed, and the study aim are now presented within the introductory section to ensure consistency with standard manuscript structure before the Methods section. (133-135)
Comments: Elaborate more about the exclusion and inclusion criteria of this study.
Response: We thank the reviewer for this valuable suggestion. In response, we have elaborated on and refined the inclusion and exclusion criteria (in sample section 153-172)
The inclusion criteria for this study targeted adults 18 years and older with a confirmed medical diagnosis of hypothyroidism for at least 6 months to reduce variability in symptom fluctuation due to recent treatment changes. The participants should be stable on thyroid medication and were receiving care in outpatient hospital clinics in the Eastern Province of Saudi Arabia. Eligible participants were required to have experienced symptoms of constipation for at least three consecutive months, consistent with the clinical definition of chronic constipation. Both Saudi citizens and non-Saudi individuals were included to ensure diversity in the sample. Additional criteria included the ability to provide informed consent, adequate cognitive capacity to complete the questionnaire, and welling to participate voluntarily.
In contrast, individuals were excluded from the study if they were younger than 18 years, patient with no prior history of constipation, and those whose constipation was attributable to non- thyroidal causes, such as irritable bowel syndrome, inflammatory bowel disorders, gastrointestinal disorders, cancer colon etc. Exclusion also applied to those who had recently undergone abdominal or pelvic surgery and were taking medications that markedly influence bowel motility (including opioids, anticholinergic, iron supplements, and certain antidepressants). Finally, individuals with severe psychiatric illness or significant cognitive impairments that could interfere with reliable data collection were not eligible to participate.
Comments: Make a section-wise detail of study tools??
Response: We thank the reviewer for this valuable suggestion. In response, we have re-write the study tool as requested ( study tool section 175-216)
The research utilized a structured questionnaire, The Bowel Habits Questionnaire, to evaluate hypothyroid patients’ knowledge, attitude, and practice regarding bowel complications associated with their condition, with a focus on constipation. The tool also evaluates the pharmacological and non-pharmacological interventions patients use to manage constipation. The questionnaire was developed by the researcher following an extensive review of the relevant literature to ensure alignment with the study objectives and existing knowledge on hypothyroidism and chronic constipation.
The questionnaire consists of four parts, each part in the questionnaire was structured with either closed-ended or multiple-choice questions or a Likert scale to gather comprehensive insights.:
- Part 1: Demographic Information, which assesses basic participants’ information (such as age, gender, duration of hypothyroidism diagnosis, and chronic disease). These variables were essential to analyze potential association between patient profiles and constipation management.
- Part 2: Participants’ Awareness of Bowel Complications Associated with Hypothyroidism: this section assessed patients' awareness about the impact of hypothyroidism on bowel function. It explored whether participants recognized constipation as a complication of hypothyroidism, whether they had discussed bowel habits with their physician, and their satisfaction with current management strategies. It also included questions on the need for additional knowledge or support.
- Part 3: Bowel Habits: this section investigated the frequency and characteristics of bowel movements, stool consistency, and constipation related symptoms. It used Likert scale questions (e.g., frequency rated from “Daily” [4] to “Never” [0]) to capture symptom severity and patterns.
- Part 4: Pharmacological and Non-pharmacological Interventions for Managing Constipation and Their Effectiveness: this section explored the strategies patients used to manage constipation. It includes pharmacological Interventions collects information on prescribed constipation medications, frequency of use, and perceived effectiveness It used Likert scale questions (rated from “Very effective” [3] to “Not effective at all [0]”. Also includes Non-Pharmacological Interventions collects information about lifestyle and dietary practices to alleviate constipation such as high-fiber diet, drinking water, and regular exercise, and other natural remedies.
To establish content validity, the tool was reviewed by a subject -matters experts. Their feedback was used to refine the questionnaire’s language, ensure clarity, and confirm the relevance and coherence of each item. Additionally, a pilot study was conducted using sample of 20 participants from the target population. This pilot testing helped assess the clarity, comprehensibility, and internal consistency of the items. Based on the pilot results, minor modifications were made to enhance the tool’s reliability and usability before proceeding with the main study (the pilot results were excluded from the research results).
Comments: Table 3 should be simpler in presentation??
Response: We appreciate the reviewer’s suggestion regarding the presentation of Table 3. We divided the table data to one small table (table 3) and one figure (figure 1) to be simple presentation. (data analysis section 316- 325)
Comments:
- Figure 1. Frequency of Constipation Management Methods Used by Participants. 264
- Make the Y-axis up to 100%.
Response: We increased Y axis up to 100%. [ figure 1 becomes figure 2 [339)
Comments: Figure 2. Perceived Effectiveness of Constipation Management Methods. 277
Please represent it in another way.
Response: We changed chart type to column 303 [figure 2 becomes figure 3, 353]
Comments:
- Table 5. Spearman’s Correlation Between Constipation Severity, Management Strategies, and Per-309?
- Could you please try to create a graphical presentation of this data?
Response: We added the graphical presentation of spearman correlation. (390)
Comments: Why does the discussion have subheadings??
Response: This is the journal guideline which is published on the website
General Improvements
In addition to addressing the specific comments, we carefully reviewed the manuscript for clarity, coherence, and formatting. Minor language edits have been made to improve readability.
We believe these revisions have substantially improved the manuscript and aligned it more closely with the aims of the Special Issue. We are grateful for the reviewers’ and editors’ insightful comments, which have guided us in enhancing the scientific rigor and practical contribution of this study.
Thank you for considering our revised submission. We hope the revised manuscript will now be suitable for publication.
With kind regards,
On behalf of all authors,

Reviewer 3 Report
Comments and Suggestions for Authors
Dears,
The following draws attention:
Methodology: The descriptive and cross-sectional design is appropriate for the study's objective. The data collection tool, a questionnaire, was validated and tested, which gives it robustness. The main limitation is the use of a convenience sample of 300 participants, which could affect the generalization of the results, especially since the ideal sample size was 380.
Discussion: The discussion is relevant and well-contextualized, as it addresses a little-researched topic in the region. The authors point out the high prevalence of constipation and the importance of patient education. The conclusions are clear and practical, recommending that nursing professionals integrate lifestyle education into care.In summary, despite the limitations of the sample, the study is valuable and its methodology is solid for a study of this kind.
Therefore, I believe they should try to see how to alleviate these limitations, although it is already recognized as a bias and how it can affect the final practical results
The objective is clearly and specifically stated, aimed at understanding the management of constipation in a defined population. However, there is a lack of precision regarding the criteria for evaluating the "effectiveness" of the interventions, which is based exclusively on the subjective perception of the participants, without clinical or physiological support.
The sample size is adequate for a descriptive study, being a non-probability sample of 300 patients with hypothyroidism in private hospitals in Saudi Arabia. The study is cross-sectional, which limits the possibility of establishing causal relationships between the variables analyzed. It is recommended to consider this limitation in the discussion, although they indicate it in the limitations section, to be less concise in the discussion, and to avoid statements that suggest causality. It would be useful to include a reflection on the representativeness of the sample. They also detail, and positively, that convenience sampling introduces selection bias, as well as the reliance on self-reporting, which is susceptible to recall bias and social desirability.
A validated instrument was used for data collection, and a pilot test was conducted to ensure reliability. Therefore, they used a validated instrument. Objective clinical data (TSH levels, intestinal transit, etc.) are lacking, which weakens the clinical interpretation of the findings. They do collect sociodemographic and clinical variables. The statistical analysis is correct and well presented. However, the effectiveness of the interventions is based on the subjective perception of the participants, without clinical or physiological support. It is suggested that objective measures be incorporated in future studies or that this limitation be more clearly recognized.
I recommend reconsideration after review as it requires substantial adjustments to strengthen its methodological rigor and interpretive clarity. The main:
1.Strengthen the discussion of methodological limitations.
2.Clarify the scope of the conclusions, avoiding unjustified extrapolations.
3.Include suggestions for future research using longitudinal or experimental designs.
4.Revise the wording in some sections to improve coherence and scientific accuracy. Regards
Author Response
Dear Reviewer,
Dear Editor,
We are grateful for the opportunity to revise our manuscript entitled “Integrated Management of Constipation in Hypothyroidism: Evaluating Pharmacological and Non-Pharmacological Interventions” (Manuscript ID: nursrep-3849959). We sincerely thank you and the reviewers for your thoughtful feedback, which has been invaluable in strengthening our work. Below, we provide a point-by-point response outlining how we have addressed each comment, and the corresponding revisions made in the manuscript.
Comments and Suggestions for Authors.
Dears,
The following draws attention:
Comments: Methodology: The descriptive and cross-sectional design is appropriate for the study's objective. The data collection tool, a questionnaire, was validated and tested, which gives it robustness. The main limitation is the use of a convenience sample of 300 participants, which could affect the generalization of the results, especially since the ideal sample size was 380.
Response: The use of a convenience sampling method and the inability to achieve a larger sample size were primarily due to practical and logistical constraints. Recruitment relied on voluntary participation through self-administered questionnaires, which inherently limits control over participant eligibility and response rates. Additionally, the strict inclusion criteria specifically requiring a confirmed diagnosis of hypothyroidism led to the exclusion of a substantial number of initial respondents (n = 82), who either did not meet this clinical criterion or provided incomplete data. These methodological choices were necessary to ensure the clinical relevance and data quality of the final sample, even though they may limit the representativeness of the broader hypothyroid population. ( results section 252 – 260)
We add this to the limitation (one notable limitation is relatively limited sample size. This reduction in sample size may limit the generalizability of the findings and reduce the statistical power to detect subtle associations or subgroup differences.). (limitation section 542=- 545)
Comments: Discussion: The discussion is relevant and well-contextualized, as it addresses a little-researched topic in the region. The authors point out the high prevalence of constipation and the importance of patient education. The conclusions are clear and practical, recommending that nursing professionals integrate lifestyle education into care. In summary, despite the limitations of the sample, the study is valuable and its methodology is solid for a study of this kind.
Therefore, I believe they should try to see how to alleviate these limitations, although it is already recognized as a bias and how it can affect the final practical results
Response: Thank you very much for your thoughtful and constructive feedback. We fully acknowledge the reviewer’s concern regarding the limitations of the sample size and the use of convenience sampling. As noted, this approach was largely driven by feasibility considerations and the strict inclusion criteria, which were necessary to ensure the reliability and clinical relevance of the study population. While these factors may have introduced some degree of bias and limited the broader generalizability of the findings, careful steps were taken to minimize their impact. For instance, clear exclusion criteria were applied to reduce potential confounding, and the analytical approach was designed to ensure internal validity.
We agree that these limitations should be transparently recognized when interpreting the results, particularly regarding their applicability to wider populations. At the same time, we believe the methodological rigor of the study, including stringent participant selection and robust analytical strategies, strengthens the credibility of the conclusions within the defined cohort. Future studies with larger, more representative samples and possibly multicenter recruitment will be important to confirm and extend these findings.
This was addressed in the limitation sections. 542-556)
Comments: This study addresses a relevant topic in the clinical field by including non-pharmacological interventions that add value to the field: the management of constipation in patients with hypothyroidism. However, there are methodological limitations and information gaps that should be considered before publication.
The objective is clearly and specifically stated, aimed at understanding the management of constipation in a defined population. However, there is a lack of precision regarding the criteria for evaluating the "effectiveness" of the interventions, which is based exclusively on the subjective perception of the participants, without clinical or physiological support.
Response: Thank you for this insightful comment. We agree that the evaluation of constipation management in our study was primarily based on participants’ self-reported perceptions rather than on objective clinical or physiological measures. This approach was intentional, as constipation is inherently a subjective symptom that is often best captured through patients’ personal experiences and reporting.
Moreover, the primary aim of our study was not to clinically validate the physiological outcomes of interventions but rather to explore hypothyroidism patients’ knowledge and practices regarding constipation, and to evaluate the perceived effectiveness of both pharmacological and non-pharmacological management strategies from the patient’s perspective. Understanding patient-reported outcomes is essential, as it provides valuable insights into real-world management, treatment adherence, and satisfaction with care factors that may not be fully reflected in clinical or physiological assessments alone.
We fully agree that the evaluation of constipation management effectiveness relies primarily on patient-reported outcomes, as constipation is largely a subjective symptom. In routine clinical practice, physicians typically consider further physiological assessments such as colonic transit studies, defecography (MRI or fluoroscopic), balloon expulsion tests, colonoscopy, or sigmoidoscopy only in more complex or refractory cases. However, since most patients in our study are covered by insurance systems, identifying participants who have undergone such advanced diagnostic procedures would be challenging, thereby limiting the feasibility of achieving an adequate sample size.
Moreover, incorporating these clinical investigations for all participants to validate their subjective reports would have required a fundamental change in the research design from a descriptive cross-sectional study to a clinical trial. This, in turn, would necessitate significantly higher funding, ethical approvals, and longer timelines, which were beyond the scope of the present study. For these reasons, our focus remained on capturing patient perspectives, which align directly with the study’s objectives. Nonetheless, we acknowledge that future research could integrate both subjective and objective assessments to provide a more comprehensive understanding of constipation management in hypothyroidism.
We did modification on the aim to be, To explore hypothyroidism patients’ knowledge and practice regarding constipation and evaluate the perceived effectiveness of pharmacological and non-pharmacological management approaches. (aim section in introducation. 135-137)
We adressed your valubale recommendations in the implication and recommendation section. 514-524
Comments: The sample size is adequate for a descriptive study, being a non-probability sample of 300 patients with hypothyroidism in private hospitals in Saudi Arabia. The study is cross-sectional, which limits the possibility of establishing causal relationships between the variables analyzed. It is recommended to consider this limitation in the discussion, although they indicate it in the limitations section, to be less concise in the discussion, and to avoid statements that suggest causality. It would be useful to include a reflection on the representativeness of the sample. They also detail, and positively, that convenience sampling introduces selection bias, as well as the reliance on self-reporting, which is susceptible to recall bias and social desirability.
Response: We addressed this valuable comment in the limitation (In addition to using convenience sampling may have introduced selection bias, potentially limiting the representativeness of the sample and, therefore, the generalizability of the results). This was addressed in the limitation sections. 542-555)
Comments: A validated instrument was used for data collection, and a pilot test was conducted to ensure reliability. Therefore, they used a validated instrument. Objective clinical data (TSH levels, intestinal transit, etc.) are lacking, which weakens the clinical interpretation of the findings. They do collect sociodemographic and clinical variables. The statistical analysis is correct and well presented. However, the effectiveness of the interventions is based on the subjective perception of the participants, without clinical or physiological support. It is suggested that objective measures be incorporated in future studies or that this limitation be more clearly recognized.
Response: Thank you for this valuable suggestion. We agree with the reviewer that the absence of objective clinical measures represents an important limitation of the present study. We have therefore emphasized this limitation more clearly in the revised discussion and highlighted the need for future research to combine both subjective and objective measures in order to validate and expand upon our findings. (limitation section 542-555))
Comments: I recommend reconsideration after review as it requires substantial adjustments to strengthen its methodological rigor and interpretive clarity. The main:
Strengthen the discussion of methodological limitations.
Response: Thank you for this helpful comment. We have strengthened the discussion of methodological limitations as suggested. Specifically, we have elaborated on the reliance on patient-reported outcomes and acknowledged the absence of objective clinical measures, noting the importance of future research that combines both subjective and objective assessments to validate and expand upon our findings. (549-555) In addition, we have highlighted the relatively limited sample size and the use of convenience sampling, which may have reduced the statistical power, introduced selection bias, and limited the generalizability of the results. These revisions have been incorporated into both the discussion and limitations sections to ensure that the methodological constraints of the study are more clearly recognized and transparently presented (542-555))
Comments: 2.Clarify the scope of the conclusions, avoiding unjustified extrapolations.
Response: The conclusions section reflected exactly what the data supports. 558-570 data is presented as insights into the participants’ specific group with a clear note for further research in the recommendation sections. In addition, the limitation was acknowledged (recommendation section 514-537
Comments: 3.Include suggestions for future research using longitudinal or experimental designs.
Response: We added this in the implication and recommendation section. (. recommendation section 514-537
Comments: 4.Revise the wording in some sections to improve coherence and scientific accuracy.
Response: Thank you for this constructive comment. In response, the entire manuscript has been carefully revised to improve coherence, clarity, and scientific accuracy. Particular attention was given to grammar, spelling, and sentence structure to ensure consistency and readability throughout the text.
We addressed the specific comments, we carefully reviewed the manuscript for clarity, coherence, and formatting. Minor language edits have been made to improve readability.
General Improvements
We believe these revisions have substantially improved the manuscript and aligned it more closely with the aims of the Special Issue. We are grateful for the reviewers’ and editors’ insightful comments, which have guided us in enhancing the scientific rigor and practical contribution of this study.
Thank you for considering our revised submission. We hope the revised manuscript will now be suitable for publication.
With kind regards,
On behalf of all authors

Round 2
Reviewer 2 Report
Comments and Suggestions for Authors
None